# Eye Health Service Uptake among People with Visual Impairment and Other Functional Difficulties in Bangladesh: A Cross-Sectional Study with Short-Term Follow Up

**DOI:** 10.3390/ijerph18179068

**Published:** 2021-08-27

**Authors:** Ruth Sanders, Ben Gascoyne, Paul Appleby, Syeda Asma Rashida, Emma Jolley

**Affiliations:** 1Sightsavers, Haywards Heath RH16 3BZ, UK; gascoynebj@gmail.com (B.G.); pappleby@sightsavers.org (P.A.); srashida@sightsavers.org (S.A.R.); ejolley@sightsavers.org (E.J.); 2Bangladesh Office-Sightsavers, Banani, Dhaka 1213, Bangladesh

**Keywords:** disability, service access, service uptake, eye health

## Abstract

People with visual impairments are disproportionately likely to also have other impairments. However, little is known about whether these other impairments affect access to eye health services. This study among cataract and refractive error patients in four districts in Bangladesh explores the relationship between self-reported difficulties in hearing, mobility, self-care, communication and cognition domains, eye health service uptake, and location of initial eye health assessment. Cataract and refractive errors were diagnosed through ophthalmic clinical assessment, and the Washington Group Short Set (WG-SS) was used to ascertain difficulties in other functional domains. Univariate and multivariate analyses were used to examine the relationship between functional difficulties and uptake of cataract and refractive error services. We found that people with self-reported functional difficulties were less likely to take up refractive error services compared to people with same need but with no functional difficulties, and that they were more than twice as likely to access surgical services after attending an outreach camp compared with a hospital facility. Since a high proportion of people attending eye health assessment services have difficulties in a range of functional domains, strategies to improve the uptake of hospital-based health services are urgently required.

## 1. Introduction

In 2020, 43.3 million people were estimated to be blind, and more than one billion others were estimated to have unaddressed refractive errors [1]. The overwhelming majority of this burden falls on communities in low- and middle-income income countries (LMICs), where many people lack access to essential services [2]. Access is unequally distributed within, as well as between, countries, and specific population groups, including women, those living in poverty, and people with disabilities, are known to be particularly vulnerable [1,3,4]. Yet relatively little is understood about the gap between the need for eye health services among people with disabilities and the level of access in LMICs [4,5]. The data that does exist suggests that people who are visually impaired, or blind, are more likely to have other impairments alongside their visual impairment, for example, hearing, communication, or physical impairments, or mental health conditions [6]. However, there remains little information about how people with visual impairments and other impairments access eye health services, and how likely they are to access services once diagnosed.

The available evidence shows that people living with disabilities are at risk of being marginalized and excluded from the social, economic, and political life of their communities, often as a result of stigma and discrimination [2]. A major systematic review of access to general healthcare services for people with disabilities in LMICs suggests a multilayered situation [4]. For example, the review’s authors found that the utilization of healthcare services and healthcare expenditure was higher for people with disabilities than people without disabilities, and people with disabilities reported experiencing greater barriers to accessing services. These barriers are believed to result from a complex combination of attitudinal beliefs, informational barriers, and practical and logistical constraints [7]. Studies from Bangladesh show that, while people with disabilities often report relatively high general healthcare needs, they can be deterred from seeking care for a range of health-system related reasons [8,9]. These include their perception of the quality of the service, the attitudes of the staff towards people with a disability, and physical accessibility, and may be mediated by personal factors such as age, religion, wealth, education, access to mobility aids, and the type of illness experienced.

In terms of how people with disabilities access eye health services specifically, knowledge is sparse. Of course, it is logical that people with visual impairments are likely to access eye care services, but what about people who have impairments in other, non-visual, domains? Data from India suggested that the proportion of self-reported difficulties, including the visual domain, among people accessing eye health services was between 15.1% and 32.3%, and a similar study in Paraguay estimated that the proportion was 27.7% [10,11]. The India study further reported that the proportion of self-reported difficulties in domains other than vision was between 7.4% and 19.1%. A recent study examined how cataract surgical coverage varied by disability status using data from five LMIC population-based surveys and found inconsistent variation in the coverage between people with and without self-reported difficulties in a number of functional domains [6]. However, the same researchers observed a significant association between visual impairment and self-reported difficulties in other functional domains across all survey sites, indicating that people with difficulties in non-visual domains were up to ten times more likely to be blind or severely visually impaired compared with others.

This highlights the importance of understanding the link between access to eye care services and impairments in other domains, such as hearing, mobility, self-care, communication, and cognition. In addition, gender is an important factor in some settings, echoing other studies that have noted the interaction between disability and gender to be ‘multiple and complex’ and linked to ‘difficult to measure’ social factors [6,12,13]. Disability, including that due to vision loss, is also highly correlated with poverty, which is an important independent predictor of barriers of access to healthcare [4].

Cataract and refractive errors are leading causes of avoidable blindness and visual impairment globally, and both are treatable with access to a well-functioning health system [14]. This is the case in Bangladesh, where the last national survey of visual impairment conducted in 2003 found that cataracts were the cause of 79.6% of blindness, and that refractive errors were the cause of 80.0% of low vision [15]. Nonetheless, ensuring equitable access to surgery or prescription spectacles in LMICs requires an accurate understanding of whether and how people with different types of impairments can access the eye health services they need. This study therefore had the following aims:To explore whether there is any association between reporting difficulties in hearing, mobility, communication, and cognition domains, and uptake of refractive error or cataract services.To understand whether the location of an initial eye health assessment (be it at a hospital facility or outreach camp) has any impact on service uptake.

## 2. Materials and Methods

### 2.1. Study Context

The data used in this study came from a project that provides eye health services at primary and secondary levels, with a particular focus on including poor and marginalized groups, including women and men with disabilities. The project operates in four of the poorest districts in Bangladesh: Sirajgonj, Rangpur, Kurigram, and Narshingdi. The project is funded by the UK government’s Foreign Commonwealth and Development Office, managed by Sightsavers, an international NGO, and implemented by four local hospitals, one based in each of the four districts. The project supports outpatient services delivered at the hospitals, as well as community-based services, known as ‘outreach camps’, which are deployed in various locations within the districts by the hospitals, using their own staff and equipment. All four hospitals are eye hospitals based in urban areas, which provide cataract surgeries, refractive error services, and other eye treatments.

For patients attending hospital facilities, there is a registration rate between 100 and 200 Taka (approx. USD 1.20–2.40). Patients who are assessed and then diagnosed and recommended for cataract surgery are given a surgery date and are expected to return later. For those who need surgery and can afford to pay, there is a charge, otherwise the costs are subsidized by the hospital and its supporting agencies.

Every month, the hospital in each district arranges eight to ten eye health outreach camps in rural communities, which are organized in collaboration with local organizations and community groups. Each camp lasts one day with the intention of reaching marginalized groups, including people with disabilities, from the surrounding areas. Before the camp, communities are informed by the local organizations of the date and location of the initial assessment. On the day, approximately seven clinical and non-clinical members of staff, including an ophthalmologist, a refractionist, and a counsellor from the hospital, attend and run the outreach camp. Any assessed cases identified for cataract surgery by the ophthalmologist are transported to the main hospital facility in a hospital vehicle for surgery. If a surgical appointment or transportation is not immediately available, arrangements are made for a vehicle to return later. At the outreach camp, registration is free of charge to the patient, and any surgery they go on to receive for cataracts is also provided free of charge.

### 2.2. Study Design

This was a cross-sectional quantitative survey of patients attending eye health services, with a short-term follow-up after three months.

### 2.3. Sample

The study participants were people over five years of age attending secondary hospital facilities and outreach camps for initial eye health assessments throughout the four districts. Participants were recruited over a three-month period between October and December 2019. The hospital facilities and outreach camps were predetermined, but patients were sampled systematically from those who attended either a hospital facility or outreach camps on days of data collection.

The sample size was designed to identify the minimum sample required to calculate a precise estimate of the disabilities in each site. This was determined using an estimated proportion of self-reported functional difficulties of 50% (to provide the largest necessary sample, given that no data were available to base this on), a confidence of 95% (z value, 1.96), and a precision or margin of error of 0.03. The minimum sample size calculated and required for each hospital facility was 382, and a further minimum sample size for outreach camps in each district was 359. Each district had a combined minimum sample size of 741, resulting in a total required minimum sample of 2964 across the study area. For each hospital facility and outreach camp, a sampling interval, X, which ranged from 1 to 10, was determined based on the average number of patients who visit per day divided by the sample size, and every Xth patient was invited to participate in the study. Data were collected for three months, and, ultimately, the busier-than-anticipated services led to the minimum sample size being exceeded in each site.

### 2.4. Data

Study participants were identified and enrolled on first contact with eye health services. Administrators at the assessment location recorded participant sex, age, and location, whether it was a hospital facility or outreach camp. At the hospital this was directly recorded in the Health Management Information System (HMIS), whereas, in the outreach camp, this was initially recorded in a paper-based form that was later input into the HMIS. Unique patient medical record numbers were also generated, which is the standard procedure for all patients.

Visual impairment, cataract, and refractive error were diagnosed by ophthalmic clinical assessment during their first contact with the service, and the clinician recorded whether a recommendation for either surgery or refractive error services was made. Ophthalmologists recommended patients for surgery when a cataract was causing visual acuity less than 6/18 or 6/12, if it was felt that it could have a detrimental effect on their livelihood. All patients identified with un or under addressed refractive errors were recommended for a prescription of spectacles. At the hospital facility, this information was entered directly into the HMIS and, at the outreach camp, this was first entered into the patient record form, and later inputted into the HMIS.

Self-reported functional difficulties were assessed using the Washington Group Short Set (WG-SS) [16]. Participants were considered to have a self-reported functional difficulty if they reported having either a lot of difficulty or that they cannot function at all in any of the hearing, mobility, self-care, communication and cognition domains. The sight domain was excluded from the analysis because the study was conducted in an eye health setting, which inflated the proportion of self-reported difficulties in this domain. Instead, the diagnostic information from patient eye screenings was considered a more accurate measure of visual impairment. The WGSS was translated into Bangla and cognitively tested before being administered by trained individuals working in each assessment location. Responses to the WG-SS was inputted into the CommCare app [17], along with the participants’ unique patient medical record number.

Uptake of refractive error and cataract surgical services was determined by reviewing the patient records from the HMIS for up to three months after the initial assessment and recommendation (until February 2020). Uptake was considered successful if the records indicated that the patient had taken up the service for which they were recommended.

### 2.5. Data Collection

Eight data collectors and one data supervisor were employed for the duration of the study. Two data collectors were assigned in each district, with one being stationed at the main hospital facility and the other at the outreach camp. Training was provided in advance of data collection, including disability sensitization, administration of the WG-SS, interview techniques, extracting data from HMIS, and using the mobile data collection app. Following this training, data collectors had some practice days to test the tools in hospital and outreach camp settings.

Responses from patients on functional difficulties, along with unique patient medical record numbers, were collected using keyless, touchscreen, ultra-mobile devices with an application designed using CommCare software (Dimagi Inc., Cambridge, MA, USA) [17]. During the enrollment data collection period, Sightsavers’ staff were responsible for reviewing and spot checking the data on a weekly basis, whilst the data supervisor provided a weekly report informing of any data collection issues or requests for technical support.

Patient enrollment and eye health status data were either automatically entered into the HMIS at the hospital facilities or were entered into a paper form at the outreach camps. At the end of the outreach camp, patient record forms were sent back to the hospital facility, and data from the forms were entered into the HMIS.

During the follow-up period, the trained data collectors accessed the HMIS and extracted the necessary data (sex, age, eye health status, location of assessment, and unique patient medical record number) by matching the unique patient medical record numbers from patient responses to the questions on functioning with treatment records in the HMIS. Due to a backlog of cataract surgeries in the project area, a three month follow-up period was introduced to allow participants time to receive their recommended treatment or cataract surgery. At the end of each month during this follow up period, the unique patient medical record numbers were updated and cross-checked with data collected at enrollment to identify which patients had been recommended for refractive error services or cataract surgery.

### 2.6. Data Analysis

This study used univariate and multivariate analyses to examine the relationship between self-reported functional difficulties and uptake of cataract and refractive error services. Chi-squared tests and logistic regression models were used to test the association between the access to cataract and refractive error services and those with self-reported functional difficulties, adjusting for several potential confounders, including age and sex. Differences in access to eye health services among those attending hospital facilities and outreach camps were also explored by including a two-way interaction term (self-reported functional difficulties with the location of the eye health services) in the regression models. Wald tests were used to determine whether the interaction was statistically significant. All analysis was conducted using Stata software version 16 [18].

### 2.7. Ethics Statement

All studies followed the fundamental principles of health research, the basis of which is to do no harm. All participants were provided with full information about the study prior to interview, along with the assurance of confidentiality. Verbal consent was requested and recorded. For anyone under the age of 18, assent alongside verbal consent from a parent/caregiver was obtained. When adults with difficulties with communication or understanding were interviewed, verbal consent was taken from an accompanying family member or carer. Where participants did not provide consent, no further questions were asked. The response rate was 99.9%, with only 3 people refusing to partake in the study. The study received ethical approval on 9 August 2018 from an independent local review board, the Centre for Injury Prevention and Research, Bangladesh (CIPRB) (Number: CIPRB/ERC/20 18/010).

## 3. Results

Overall, 4136 patients were enrolled in the study, and Table 1 presents the characteristics of the sample. The total number of participants exceeded the original estimated minimum sample required by approximately 40%, evenly distributed across all sites. This excess in enrolment was due to services being busier than anticipated and the minimum sample size being reached quickly. Despite this, data collectors continued collecting data until the end of the assigned three-month data collection period, resulting in a much larger sample size.

A slightly higher proportion (53.3%) were female, and sample females were younger than males. More than half the sample (2302; 55.7%) were assessed at a hospital facility as opposed to an outreach camp. Of the participants, 833 (20.1%) were identified as having cataracts (single or bilateral) and were recommended for surgery, and 1809 (43.7%) were recommended for refractive error services.

The proportion of self-reported functional difficulties in the hearing, mobility, self-care, communication, and cognition domains among all participants was 26.6% (see Table 2). There was a statistically significant gender gap of almost ten percentage points (31.4% for women, 21.1% for men).

Among individual domains of functioning, there were significant gender differences in self-reported difficulties in walking and cognition. Moreover, 44.7% of women reported mobility difficulties, compared to 33.1% of men, and the proportion of sampled women who had difficulties with cognition was also higher than the proportion of men (56.9% and 46.4%, respectively).

The proportion of self-reported functional difficulties in the sample was significantly higher among people assessed at outreach camps compared with the group assessed at hospital facilities (34.4% and 20.4%, respectively). People assessed at outreach camps were also older, and slightly more were female (see Table 3).

Results from the multivariate logistic regression presented in Table 4 indicate a significant negative association between self-reported functional difficulties and access to refractive error services. Specifically, the odds of people with a self-reported functional difficulty accessing refractive errors services were lower than for the people without a self-reported functional difficulty requiring the same services, after adjusting for sex, age, and location (OR 0.68, 95% CI 0.53,0.89, *p* = 0.005). No significant association was observed between self-reported functional difficulties and the uptake of cataract surgical services (*p* = 0.345).

A subgroup analysis examined whether access to eye care services differed by self-reported functional difficulty status and location of assessment. Results from the two-way interaction model (self-reported functional difficulties with location of assessment) presented in Table 5 show that the predicted odds of accessing cataract surgical treatments for people who self-reported experiencing functional difficulties depended on the initial location of where they were assessed (OR 2.12, 95% CI 1.04,4.32, *p* = 0.038). That is, people with cataracts and self-reported functional difficulties were more than twice as likely to access surgical services after having been assessed at an outreach camp than those who were initially assessed at a hospital facility. The significance of this interaction was confirmed by a Wald test ($\chi^{2}$ = 4.32, *p* = 0.038). There was no meaningful interaction between access to refractive error services, self-reported functional difficulties, and location of assessment (*p* = 0.249).

## 4. Discussion

The primary aim of this study was to examine the associations between self-reported functional difficulties and uptake of recommended cataract surgery and refractive error services using data collected as part of a Sightsavers’ program in Bangladesh. The results show that 26.6% of people attending eye health assessment services had self-reported functional difficulties. In terms of eye health service uptake, this study found that people with self-reported functional difficulties were less likely to access refractive error services compared to people with the same need but with no functional difficulties. Although no difference was observed in cataract surgical uptake between people with and without self-reported functional difficulties, differences were noted among those with self-reported functional difficulties by location, where uptake was more than twice as likely among those who were initially assessed at an outreach camp compared with those who were initially assessed at a hospital facility.

The sample proportion of self-reported functional difficulties in this study was noticeably higher than in other studies in Bangladesh [9,19,20]. The proportion is also higher than the results of the studies from eye health services in India and Paraguay [10,11]. The higher burden of self-reported functional difficulties among women was also similar to that observed in Bangladesh and India [11,19]. This observed higher proportion may be attributable to the emphasis placed on the eye health services to reach particularly marginalized groups of people, including women with self-reported functional difficulties, and the specific mobilization techniques used to attract people to the services. A population-based study on disability among children in Bangladesh found significant differences between overall impairments at assessment centers (9.0%) compared with those at a household level (14.7%) [21]. Although the study had lower prevalence estimates, it emphasizes that comparable population-based data is needed to understand the eye health needs of people with additional impairments, and that, without it, is not possible to say whether the services in this study are reaching all the people who need treatment. However, the 26% of people experiencing self-reported functional difficulties observed in the present study is broadly comparable to the 22% of adults with disabilities reported by Mitra and Sambamoorthi, who analyzed responses to four questions from the WG-SS (seeing, moving, remembering, and self-care) in the 2002–2004 World Health Survey in Bangladesh [22]. The high proportion of self-reported functional difficulties observed in this study underscores the importance of disaggregating routine eye health data by sex and disability, which can be used to ensure that eye health services are accessible for all men and women with and without disabilities.

While these data show that, at least to some extent, people with additional impairments are attending eye health services, they also suggest that there are inequalities of access to services, and although it cannot provide explanations, several possibilities exist. Finances are often a barrier for people accessing health services, particularly among those with a disability [2]. A 2017 systematic review of poverty and disability in low- and middle-income countries found strong evidence to support the link between disability and economic poverty, with 122 of 150 (81%) of included studies reporting a positive relationship between the two variables [2]. In relation to eye health service provision, the combination of direct costs, such as the cost of spectacles or cost of surgery, and indirect costs, such as transport, accommodation, and food, are known to act as a significant deterrent for people with disabilities [23]. Although the project in this study heavily subsidized cataract surgeries, it did not address the cost of spectacles. Glasses are comparatively expensive in Bangladesh, and can cost as much as 600 taka (USD 8), which is approximately one and a half times the average daily wage; and patients from lower income households, which often includes people with additional impairments, are often less willing to pay for these services [24].

This study also observed that access to cataract surgery for people with self-reported functional difficulties was significantly associated with the initial assessment location. Participants with self-reported functional difficulties were more than twice as likely to access surgical services when they attended assessment at an outreach camp than at a hospital facility. This result is similar to that observed in the India study, where the proportion of self-reported functional difficulties was more than four-times higher among patients attending community-based services (36.4%) than those attending hospitals (7.5%) [11]. In Bangladesh, this might be partially explained by the outreach camp model used within the project area targeting those who are more likely to be excluded from services, whereby free transport to and from the hospital facility was provided to cataract patients after assessment, as was the cost of their cataract surgery and any overnight stay at a hospital facility. However, those screened at the hospital facility who needed cataract surgery but were not eligible for subsidized treatment, or were unaware of the eligibility criteria, were expected to pay a fee. In addition to the cost of surgery, the difficulty of travelling to a faraway hospital may be a deterrent to some patients who require a family member to accompany them, whether for accessibility reasons in the case of people with certain types of impairments, but also for social reasons in the case of particularly older women [25]. Outreach camps, particularly where transport to the surgical services is provided, may overcome accessibility issues associated with both disability and gender that have been identified in other rural settings [19,26]. Physical accessibility to hospital facilities could still be an issue, despite the infrastructure improvements that were made after conducting accessibility audits, and may account for differences in service uptake. These findings suggest that core project activities such as outreach camps located within rural communities, transport provision, and fee subsidies are having their intended effect of reaching marginalized individuals, however questions of sustainability after the end of the project remain.

This study also highlights the importance of geographic location in relation to service uptake for people with a range of impairments in Bangladesh. A situational analysis conducted by the project in 2019 found that the distance to hospital facilities was a major barrier for patients that resulted in a reluctance to travel for primary eye health care [27]. The outreach camp model used by the project demonstrates the high level of demand for services from people with disabilities in more rural areas. However, the sustainability of this model is not understood and, without a long-term approach to provide consistent accessible eye health services in rural areas, it is reasonable to expect that people with disabilities may continue to experience barriers to access based on their geographic location. This observation is consistent with a study completed by Eide et al., which indicated that the limited availability of services in rural areas and the poor availability of transportation were frequently perceived by people with disabilities as significant barriers to access [28]. These are important factors for the project, as they consider the sustainability of outreach camps and transport provision, all the while ensuring that eye health services are accessible, particularly for women and people with disabilities.

There were some limitations to this study. Firstly, data did not include participant home locations, economic status, or what they paid for their services, which are understood to be key determinants of access to eye health services. There are no population data in the study areas available on disability and visual impairment with which to compare service level data and understand the extent of exclusion. The project has been running since July 2018, and we are unable to assess the extent to which project activities designed to increase access to health services among people with disabilities, particularly those with additional impairments, have affected the number of people accessing services. Finally, there was a backlog of surgeries, and many patients who had been identified with cataracts and recommended for surgery may not have received their surgery over the data collection period and are therefore not accounted for in the data.

## 5. Conclusions

This study examined the associations between self-reported functional difficulties and eye health service uptake among patients in four districts in Bangladesh. It showed that there was a high proportion of people with self-reported functional difficulties among those accessing eye health services, and that people with other impairments appear to be disadvantaged in terms of their uptake of certain services. Furthermore, the location of the initial eye health assessment did have an impact on service uptake, wherein people reporting self-reported functional difficulties were more than twice as likely to access cataract surgical services if they had been assessed at an outreach camp compared with a hospital facility. This could suggest that some inequalities may be reduced if targeted and if comprehensive services are provided in rural areas where certain barriers such as finances and transport can be removed. However, rigorous evidence on the interventions and strategies that improve the uptake of services among disadvantaged groups, such as people with disabilities, is extremely limited, and further investment and research is required in this area.

## Figures and Tables

**Table 1 ijerph-18-09068-t001:** Sample characteristics of all participants assessed at eye services in four districts between October and December 2019.

Characteristics	Total		Males		Females		*p*-Value *
	*n*	%	*n*	%	*n*	%	
Sex							
Male	1931	46.7					
Female	2205	53.3					
Age							
<18 years	373	9.0	178	9.2	195	8.8	<0.001
18–49 years	1783	43.1	778	40.3	1005	45.6	
50–69 years	1479	35.8	652	33.8	827	37.5	
>69 years	501	12.1	323	16.7	178	8.1	
Location of assessment							
Hospital facility	2302	55.7	1135	58.8	1167	52.9	<0.001
Outreach camp	1834	44.3	796	41.2	1038	47.1	
Recommended for cataract services							
No	3303	79.9	1524	78.9	1779	80.7	0.160
Yes	833	20.1	407	21.1	426	19.3	
Recommended for refractive error services							
No	2327	56.3	1176	60.9	1151	52.2	<0.001
Yes	1809	43.7	775	39.1	1054	47.8	

* *p*-value based on Chi-squared test.

**Table 2 ijerph-18-09068-t002:** Sample proportion, domain, and severity of self-reported functional difficulties among all participants assessed in four districts.

Domains of Functioning	Severity of Functional Difficulty	Total		Males		Females		*p*-Value *
		*n*	%	*n*	%	*n*	%	
Self-reported functional difficulties ^1^	No	3035	73.4	1523	78.9	1512	68.6	<0.001
	Yes	1101	26.6	408	21.1	693	31.4	
Hearing	Any difficulty	1069	25.9	446	23.1	623	28.3	0.657
	Some difficulty	855	80.0	351	78.7	504	80.9	
	A lot of difficulty	197	18.4	87	19.5	110	17.7	
	Cannot do at all	17	1.6	8	1.8	9	1.4	
Mobility	Any difficulty	1625	39.3	640	33.1	985	44.7	0.002
	Some difficulty	970	59.7	416	65.0	554	56.2	
	A lot of difficulty	641	39.5	219	34.2	422	42.8	
	Cannot do at all	14	0.9	5	0.8	9	0.9	
Cognition	Any difficulty	2149	52.0	895	46.4	1254	56.9	<0.001
	Some difficulty	1610	74.9	713	79.7	897	71.5	
	A lot of difficulty	508	23.6	169	18.9	339	27.0	
	Cannot do at all	31	1.4	13	1.5	18	1.4	
Self-care	Any difficulty	367	8.9	128	6.6	239	10.8	0.718
	Some difficulty	244	66.5	87	68.0	157	65.7	
	A lot of difficulty	103	28.1	33	25.8	70	29.3	
	Cannot do at all	20	5.5	8	6.3	12	5.0	
Communicating	Any difficulty	301	7.3	124	6.4	177	8.0	0.099
	Some difficulty	217	72.1	85	68.6	132	74.6	
	A lot of difficulty	71	23.6	30	24.2	41	23.2	
	Cannot do at all	13	4.3	9	7.3	4	2.3	

^1^ Based on self-reported difficulties in five domains * *p*-value based on Chi-squared test.

**Table 3 ijerph-18-09068-t003:** Self-reported functional difficulties, sex, and age characteristics by location of assessment.

Characteristics	Hospital Facility (*n* = 2302)		Outreach Camp (*n* = 1834)		*p*-Value
	*n*	%	*n*	%	
Self-reported functional difficulties					
No	1832	79.6	1203	65.6	<0.001
Yes	470	20.4	631	34.4	
Sex					
Male	1135	49.3	796	43.4	<0.001
Female	1167	50.7	1038	56.6	
Age					
<18 years	276	12.0	97	5.3	<0.001
18–49 years	1185	51.9	598	32.6	
50–69 years	671	29.2	808	44.1	
>69 years	170	7.4	331	18.1	

**Table 4 ijerph-18-09068-t004:** Results of logistic regression models showing an association between self-reported functional difficulties and eye health service uptake among participants recommended for cataract and refractive error services.

Characteristics	Cataract Services (*n* = 833)			Refractive Error Services (*n* = 1809)		
	OR	95% CI	*p*-Value	OR	95% CI	*p*-Value
Self-reported functional difficulties						
No						
Yes	0.87	0.64, 1.17	0.345	0.68	0.53, 0.89	0.005
Sex						
Male						
Female	0.99	0.73, 1.33	0.937	1.17	0.93, 1.47	0.176
Age						
<18 years						
18–49 years	3.79	0.37, 38.63	0.260	0.95	0.65, 1.38	0.784
50–69 years	6.26	0.64, 61.22	0.115	0.93	0.62, 1.39	0.711
>69 years	5.84	0.59, 57.54	0.130	0.76	0.43, 1.37	0.367
Location of assessment						
Hospital facility						
Outreach camp	1.67	1.17, 2.37	0.004	0.65	0.51, 0.82	<0.001

**Table 5 ijerph-18-09068-t005:** Results of interaction model showing association between self-reported functional difficulties, location of assessment, and eye health service uptake among participants recommended for cataract and refractive error services.

Characteristics	Cataract Services (*n* = 833)			Refractive Error Services (*n* = 1809)		
	OR	95% CI	*p*-Value	OR	95% CI	*p*-Value
Self-reported functional difficulties						
No						
Yes	0.48	0.26, 0.91	0.023	0.78	0.56, 1.10	0.156
Sex						
Male						
Female	0.98	0.73, 1.32	0.910	1.16	0.93, 1.46	0.194
Age						
<18 years						
18–49 years	3.98	0.39, 40.30	0.242	0.94	0.64, 1.37	0.729
50–69 years	6.54	0.67, 61.47	0.106	0.91	0.61, 1.37	0.662
>69 years	6.13	0.63, 59.94	0.119	0.75	0.42, 1.35	0.342
Location of assessment						
Hospital facility						
Outreach camp	1.21	0.75, 1.93	0.436	0.71	0.53, 0.94	0.019
Interaction ^1^	2.12	1.04, 4.32	0.038	0.74	0.44, 1.24	0.249

^1^ Two-way interaction term (self-reported functional difficulties with outreach camp).

## Data Availability

Data presented in this study are available on reasonable request from the corresponding Author. The data are not publicly available due to information that could compromise the privacy of research participants.

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
