# Peer review of "Eye Health Service Uptake among People with Visual Impairment and Other Functional Difficulties in Bangladesh: A Cross-Sectional Study with Short-Term Follow Up"

_ijerph, 2021, doi:10.3390/ijerph18179068_

Round 1

Reviewer 1 Report

The authors studied the prevalence of eye health care services in volunteers with physical disabilities. An interesting finding is that people with physical disabilities are more likely to have access to cataract services. However, there are some concerns about the definitions of words and phrases in this study.

Major:

  1. What are additional non-visual disabilities? The authors defined the additional disabilities at lines 35 – 37. I recognized that the meaning of additional disabilities in this study was defined as the combination of vision disability and other disabilities. However, the additional non-visual disabilities did not define in the main text. The definition needs to be clarified as to whether the author has visual impairment only or other disabilities without visual impairment as well. Furthermore, the comma makes it hard to understand the word “additional non-visual disabilities”.

  1. How prevent the cataract and refractive error? The prevention methods for these diseases have not been established. If cataract surgery can prevent blindness, I recommend it be clearly stated. The authors would like to say that blindness is caused by high myopia, but there is no way to stop the progression of myopia completely. Moreover, the information on the evidence of vision loss is too old.

  1. The aim of this study is unclear. What is the difference between non-visual disabilities and “additional” non-visual disabilities?

  1. I recommend that correspond to the P-values in the Tables. I think that the authors statistically analyzed between males and females of all data. The current P-values locate at “Cannot do at all”.

Minor:

  1. Line 28, typo. “. [1]. “
  2. Reference does not consist of journal format. e.g., 23: PLOS ONE, 25: PLos One. I recommend that the authors carefully check the reference list.

Reviewer 2 Report

Summary

It seems as if this study is trying to do too many things:

  1. To compare the proportion of people who attend outreach screening camps, or who present to an eye hospital, who are visually impaired from cataract and refractive errors who do or do not have an additional disability.
  2. To compare the uptake of referral and acceptance of treatment for cataract and refractive error amongst those identified in the outreach screening camps who were visually impaired from these conditions who did and did not have an additional disability. [The flow diagram at the end relates to this].

The findings from 1. are not as interesting or informative as for 2.

To explore 2 the sample size needs to be large enough to be able to compare people with VI from cataract and RE with or without additional disabilities. We are not shown the data. Presumably one of the reasons for the lack of a statistically significant difference for cataract was because the sample size was too small?

Terminology

It is very confusing to talk about screening in a hospital setting, where those attending will have a far more detailed examination than just a screening test.

It is also confusing to talk about referral within a hospital; maybe cataract surgery or refraction for refractive error were recommended?

Using different terms for the different locations will help, as the study is currently very difficult to understand.

Major omission

How was an additional disability defined? Did community participants need to be visually impaired (define the level of vision) AND score in the top 2 categories of one or more of the 5 Washington Group domains? This is not clear. Presumably the study only focused on people with vision impairment from cataract and refractive errors, as these data could be readily accessed from the hospital HMIS?

Introduction

Line 79. This sentence reads as if cataract and refractive errors are preventable, which they are not -  the visual impairment from these conditions is treatable.

More information is needed on the outreach screening camps, particularly as in the discussion we read that emphasis was put on increasing access by marginalized groups.

Who examines those who attend the screening camps? The reader needs to know their level of skill and the likely accuracy of their reason for referral.

Aim: as this is a facility and eye camp based study it is not accurate to talk about the prevalence of a condition; the word proportion is more appropriate. Please change throughout the text.

How many camps were held during the study period? What was the total number of people who attended?

Methods

A flow diagram would help (see attachment).

Line 120: They were not selected using simple random sampling, which is what the words “random sampling” suggest, as they were sampled systematically, using a fixed interval.

Line 125: it is not clear whether the sample size of 2,964 refers to the total for all 8 sites, or a total per district, or the total for the 4 hospitals and for the 4 camps. From my reading of the results it seems as if this is the total sample. Why were there so many more in the hospital component compared with the camps? The number of participants is very different from the 2,964 sample size…

Line 146; it would be clearer to say “….location of eye hospital and screening camp” as screening is will not be taking place in the hospital.

From reading to the end of the methods it seems as if the surveys were undertaken only in the camps, and the attendance of those referred was checked using the hospital HMIS. If this is the case then the aim 1 needs to be revisited. From my understanding of the methods, the surveys in the camps identified those who needed hospital services whether they had an additional disability or not (see below). Hospital records were cross checked to see whether they had attended the hospital and undergone treatment. If this is the case, I do not understand line 120/21 “patients were randomly sampled from those who attended on the assigned days of data collection.”

Line 181: I am now very confused! What is the difference between a camp and a hospital screening location”?

Data analysis:

What are the “key outcomes of interest”? These need to be explained. From my reading it seems as if the main outcome should be the uptake of referral and then treatment for cataract and refractive error by people with visual impairment who did or did not have an additional disability.

Line 166: does the camp data system send data on all those who need hospital care to the hospital the same day? This is not very clear and needs to be explained.

Line 150. Drop the word “up”

Results

I find the results difficult to interpret as I do not follow the methods and too few key pieces of data are provided.

Table: Do these data refer to people visually impaired from cataract and uncorrected refractive errors? Not clear.

Table 4: More data are needed before these analyses are presented on the numerators and denominators used in this analysis so that the reader understands what the analyses are based on. The heading needs to be more specific so the reader knows that this refers to people with VI from uncorrected refractive errors only.

Discussion

I have not read this as i do not understand the methods and not enough detail is provided in the results.

Smaller points

Data are plural – should say data were (line 118, 148 etc)

Round 2

Reviewer 1 Report

The authors have made efforts to revise the paper. This paper almost satisfies with the publication. I recommend a small revise before publication.

Line 281-284. The authors write the reasons that the actual sample size was 40% larger than predicted. What is implied by writing this information? There is no mention of the relevance of the fact that the data aggregator was employed for three months.

Reviewer 2 Report

The revised version of this paper is now much clearer.

There are a few typos, and the pluarility of data needs to be adddressed in several places

Line 41  “Data that do…

Line 114  “data…come” form.

Line 158: “data were”.

Line 133, “data … were entered”.

Line 446 “there are no population”…

Typo: Line 271 change basic to basis

Typo: Line 300. Add “in” walking
